# Generalist large language models in a specialized world: Evidence from the Italian national medical education pathway

**Tommaso Mario Buonocore**[1]*, **Antonio Russo**[2], **Dario Mingarelli**[2], **Enea Parimbelli**[1,3]

**1** Department of Electrical, Computer and Biomedical Engineering, University of Pavia, Pavia, Italy,
**2** R&D and Operations Departments - Consorzio Interuniversitario CINECA, Bologna, Italy, **3** Telfer
School of Management, University of Ottawa, Ottawa, Canada

\* tommaso.buonocore@unipv.it

## Abstract

Creating language-specific and domain-specific large language models presents substantial challenges, including computational demands and limited data availability. While it is commonly believed that the benefits of specialized models justify these challenges, we dispute this notion with a comparative evaluation in a low-resourced language and medical-specific domain. In our study, we analyze the performance of various LLMs applied to the Italian healthcare domain using novel unpublished datasets, consisting of five-choice questions from national pre-university and post-university medical exams, covering clinical and preclinical fields. As part of this work, we release these datasets to the research community. We evaluated multilingual and Italian-specific models, along with general-purpose and healthcare-specific models, spanning both open-source and proprietary architectures of varying sizes. Our results demonstrate that multilingual, general-purpose large models consistently exceed the pass threshold across all tests, with the best models achieving over 90% accuracy on postgraduate-level exams. Model size emerged as the most critical factor influencing performance, whereas domain specialization and single-language localization offered no evidence of specialization superiority. These findings challenge the traditional pretrain-then-finetune paradigm for domain and language localization in language models, suggesting that advancements in generic-purpose multilingual models may render domain-specific pretraining unnecessary in many specialized cases.

## Author summary

Artificial intelligence researchers have long assumed that to perform well in a specific field or language, AI models need to be specially trained on relevant data. This study examines that assumption using Italian medical exams as a benchmark. We evaluated 37 AI language models across a wide range: some built specifically for medicine, some optimized for Italian, and others

**Data availability statement:** The datasets analyzed during the current study are publicly available at the following Huggingface repository collection: hf.co/collections/Detsutut/italian-national-medical-education-pathway-tests. To mitigate the risk of automated scraping by LLMs, which could compromise the integrity of future model evaluations, access is granted only after users manually acknowledge an agreement not to use the data for model training without explicit disclosure. This procedure ensures compliance with data leakage prevention measures.

**Funding:** This work is funded by the "Hub Life Science – Digital Health (LSH-DH) PNCE3-2022-23683267 - DHEAL-COM Project – CUP: F13C22002030001", funded by the Italian Ministry of Health as part of the National Complementary Plan Innovative Health Ecosystem - CUI: 20 PNC-E.3. This publication reflects only the authors' view and the Italian Ministry of Health is not responsible for any use that may be made of the information it contains.

**Competing interests:** The authors have declared that no competing interests exist.

general-purpose tools designed to work across many topics and languages. All were tested on previously unpublished test from official Italian medical exams, used to assess aspiring doctors at different stages of their training, from medical school admission through specialization. The results suggest that neither medical specialization nor language fine-tuning provided a consistent advantage for this task. Model size emerged as the strongest predictor of performance, with the largest multilingual general-purpose models not only exceeding the passing threshold but also surpassing average human candidate scores on most exams. For healthcare institutions considering AI adoption, this points toward a need to carefully evaluate whether the costs of building and maintaining specialized models are justified. Given the current capabilities of general-purpose systems, resources may be better directed toward clinical validation and safety testing, integration into real-world workflows, data infrastructure and governance, and ensuring that clinicians can interpret, audit, and appropriately trust AI outputs.

## 1. Introduction

The emergence of large language models (LLMs) has fundamentally transformed our approach to artificial intelligence in healthcare, challenging long-held assumptions about the necessity of specialized rule-based systems. Since the introduction of BERT in 2018 by Delvin et al. [1], the medical AI community has witnessed a proliferation of domain-specific models, from BioBERT's pioneering work in biomedical text mining [2] to specialized generative transformers for clinical applications like BioGPT [3], FHIR-GPT [4] or Meditron [5]. From the linguistic point of view, monolingual corpus and monolingual models have been derived from the original base model for thousands of languages and dialects.

This trend towards domain and language specialization reflects a broader paradigm in machine learning: the belief that models trained on domain-specific and language-specific data invariably outperform their general-purpose, multilingual counterparts. Indeed, many studies have demonstrated impressive results, with specialized models achieving superior performance in tasks ranging from financial text classification to biomedical entity recognition [6]. However, the advent of foundation models, characterized by their massive scale and zero-shot learning capabilities, has unequivocally begun to challenge the conventional wisdom of both domain and language specialization, suggesting that general, versatile LLMs may possess emergent capabilities that enable them to adapt to specialized domains without explicit training [7]. The cross-domain/cross-lingual transfer phenomenon has been particularly evident in the medical context, where foundational multilingual models like GPT-4 have demonstrated remarkable capabilities across languages and specialized medical tasks despite no explicit medical training [8–10].

The Italian healthcare system provides a fascinating natural experiment for exploring further these dynamics. As one of Europe's oldest and most comprehensive

healthcare systems, it combines rigorous academic traditions with modern medical practice, reflected in its stringent examination systems for medical professionals. Previous attempts to develop Italian-specific medical AI systems have followed the traditional specialization approach, with projects like BioBIT [11] and Igea [12] focusing on collecting Italian biomedical texts for domain-and-language-specific fine-tuning. However, these efforts have faced significant challenges, including the limited availability of Italian medical corpora and the computational costs of developing specialized models in a rapidly evolving field. Model specialization becomes particularly relevant when considering the economic and practical implications for healthcare systems. Training and maintaining specialized models for each language and medical subdomain requires substantial resources: a luxury only a few healthcare institutions worldwide can afford.

In this study, we systematically examine these questions through the lens of Italian medical education and practice. By leveraging previously unpublished datasets derived from national medical examinations, we provide a unique window into the comparative performance of specialized and general-purpose models in a high-stakes medical context. Our investigation spans different medical fields and all levels of the educational pathway, from medical school admission to licensing and specialization examinations, capturing the full spectrum of knowledge required to become a physician in the Italian healthcare system. This comprehensive coverage allows us to assess not only overall model performance but also whether the incremental expertise expected of human learners is reflected in LLM behavior.

Our study further distinguishes itself from prior research like Alfertshofer et al. [9] and Riedemann et al. [13] by evaluating multiple model families rather than focusing solely on GPT-based systems, enabling a broader and more balanced view of model capabilities across the physician's educational trajectory. Moreover, the use of previously unpublished datasets ensures that the test items are absent from model training corpora, mitigating concerns regarding data leakage or overfitting.

Finally, we complement model results with human performance data directly obtained from national reports covering the entire target population, enabling a population-level comparison that situates LLM performance in the real-world context of medical education.

## 2. Methods

The study was conducted and reported following the TRIPOD-LLM guidelines for studies using large language models [14]. The completed checklist is provided in Table A in S1 Text. The source code is shared (github.com/detsutut/llm-evaluation/) to guarantee reproducibility.

### Large language models

Large Language Models (LLMs) are advanced neural networks designed to process and generate human-like text based on vast amounts of data. These models, such as GPT and Llama, leverage billions of parameters to perform complex tasks like translation, summarization, and question-answering. By understanding patterns and context, LLMs enable breakthroughs in natural language understanding and generation across various industries, including healthcare [15].

**Selection criteria.** Due to the increasing availability of GPU for accelerated training and inference, as well as optimization tools and no-code frameworks, developing LLMs is becoming more and more affordable for small companies and individuals without the need for high-end, expensive computational power. At the time of writing, the HuggingFace repository, a reference point for the open-source LLM community, reports 1172925 models, 191117 of them designed for text and text-to-text generation. Among these, 2521 are reported as compatible with the Italian language, while 38736 are intended for English. This extremely repository has been further expanded including also the major players providing closed-source models, namely Anthropic AI, Open AI, and Google DeepMind.

Models have been then filtered using the criteria reported in Table 1. While selective, these criteria still lead to a number of candidates (more than 1000) hardly manageable in an empirical evaluation, thus requiring further filtering. Therefore, additional selection criteria have been applied by consulting the open Italian (general) and International (medical)

**Table 1. Selection Criteria.**

- Cut-off date: August 2024
- Source: open source, open weights, private
- Language: Italian, multilingual including Italian
- Domain: generic, medical. Models aligned for a specific medical subdomain (e.g., radiology, neurology) have not been included
- Task: task-agnostic models only. Models explicitly fine-tuned to solve a single task, such as closed-book QA, have been excluded
- Version: most recent update (i.e., Claude Sonnet 3.5 over 3), instruction-tuned if multiple versions are available (i.e., Gemma 2 9B IT over Gemma 2 9B)
- Sizes: above 0.5 B. For mixture of experts, only the active parameters at inference time will be counted.
- Accessibility: the model should be either accessible through API or downloadable for local inference. Due to computational limitations, models above 70 billion parameters have been considered only if API-accessible.

leaderboards on Hugging Face (openlifescienceai/open_medical_llm_leaderboard, mii-llm/open_ita_llm_leaderboard) and then by picking the best-performing models in terms of average accuracy for each category of parameter size, using the same cut-off date described above.

This further refinement leaves us with a final list of 37 candidate models developed by 19 organizations or individuals, covering four orders of magnitude regarding active parameter count. The majority of these models are open-source or open-weights (29/37), general-purpose (27/37), or multilingual (27/37), as reported in Table 2. Details about repositories and endpoints for each model are reported in Table B1 and B2 in S1 Text.

## Data

The performance of the candidate LLMs has been measured over novel unpublished question-answering datasets, specifically designed to assess medical knowledge on various levels. Each dataset consists of five-choice questions, sourced from national pre-university and post-university medical exams, covering both clinical and preclinical fields. In particular, three datasets have been collected for evaluation.

*ITMSA (ITalian Medical School Admission test).* This dataset includes 3301 undergraduate-level questions sourced from national placement tests required to access the 6-year-long Medical School in Italian universities. These questions are divided into 5 topics: Biology, Chemistry, General Knowledge, Maths & Physics, and Logic. Created in July 2024.

*ITMSA (ITalian Medical School Admission test).* This dataset includes 3301 undergraduate-level questions sourced from national placement tests required to access the 6-year-long Medical School in Italian universities. These questions are divided into 5 topics: Biology, Chemistry, General Knowledge, Maths & Physics, and Logic. Created in July 2024.

*ITMLE (ITalian Medical Licensing Examination).* This dataset includes 6696 postgraduate-level questions sourced from the national medical licensing examination necessary to practice the medical profession. These questions are split into two sub-areas. The clinical area includes 3390 questions related to the clinical practice, spanning over 9 different topics: General Surgery, Specialized Surgery, Internal Medicine, Legal Medicine and Public Health, Specialized Medicine, Sensory Organs, Obstetrics and Gynecology, Pediatrics, Radiology, and Diagnostics; the preclinical area consists of 3306 questions related to the preclinical practice, spanning over 5 different topics: Pharmacology, Physiology, Morphology, Pathology, Prevention and Medical Ethics. Created in February 2020.

*ITMRA (ITalian Medical Residency Admission test).* This dataset includes 966 postgraduate-level questions sourced from the national placement tests required to access medical residency in Italian hospitals. Admission tests span from November 2017 to July 2023.

Table 2. **List of all the models included in the study.** † Undisclosed size, approximated value reported based on cost-performance estimates, tech reports, and interviews.

| Name | Active Params | Company | Language | Medical | Instruct | Open |
|---|---|---|---|---|---|---|
| Qwen 2 1.5B Instruct [16] | 1.5 | Alibaba | Multi | | × | × |
| Qwen 2 7B Instruct [16] | 7 | Alibaba | Multi | | × | × |
| Claude 3 Haiku | >50† | Anthropic | Multi | | × | |
| Claude 3.5 Sonnet | >100† | Anthropic | Multi | | × | |
| Claude 3 Opus | >500† | Anthropic | Multi | | × | |
| Igea 1B Instruct [12] | 1 | BMI Lab Unipv | Ita | × | × | |
| Igea 3B Instruct [12] | 3 | BMI Lab Unipv | Ita | × | × | × |
| Apollo 0.5B [17] | 0.5 | CUHKSZ NLP | Multi | × | | × |
| Apollo 2B [17] | 2 | CUHKSZ NLP | Multi | × | | × |
| Apollo 7B [17] | 7 | CUHKSZ NLP | Multi | × | | × |
| Meditron 3 8B [18] | 8 | EPFL & co. | Multi | × | | × |
| Meditron 3 70B [18] | 70 | EPFL & co. | Multi | × | | × |
| Gemma 2 2B IT [19] | 2 | Google | Multi | | × | × |
| Gemma 2 9B IT [19] | 9 | Google | Multi | | × | × |
| Gemini 1.5 Flash | >30† | Google | Multi | | × | |
| Gemini 1.5 Flash 8B | 8 | Google | Multi | | × | |
| EMO 2B | 2 | HelpingAI | Multi | | | × |
| Modello Italia 9B | 9 | iGenius | Ita | | | × |
| JSL MedPhi 2 2.7B | 2.7 | John Snow Labs | Multi | × | | × |
| MedLlama 3 | 3 | MAILAB Yonsei | Multi | × | | × |
| Llama 3.1 8B Instruct [20] | 8 | Meta | Multi | | × | × |
| Llama 3.1 70B Instruct [20] | 70 | Meta | Multi | | × | × |
| Llama 3.1 405B Instruct [20] | 405 | Meta | Multi | | × | × |
| Phi 3.5 mini instruct [21] | 3.8 | Microsoft | Multi | | × | × |
| Phi 3 medium 4k instruct [21] | 14 | Microsoft | Multi | | × | × |
| Mistral 7B Instruct [22] | 7 | Mistral | Multi | | × | × |
| Mixstral 8x7B Instruct [23] | 7 | Mistral | Multi | | × | × |
| Mistral Large 2 | 123 | Mistral | Multi | | | × |
| Llama 3.1 8B Ita | 8 | No (Individual) | Ita | | | × |
| Qwen 2 1.5 ITA Instruct | 1.5 | No (Individual) | Ita | | × | × |
| Llama MedX | 8 | No (Individual) | Multi | × | | × |
| GPT 4o | >300† | OpenAI | Multi | | × | |
| GPT 4o mini | >50† | OpenAI | Multi | | × | |
| Minerva 1B Base | 1 | Sapienza NLP | Ita | | | × |
| Minerva 3B Base | 3 | Sapienza NLP | Ita | | | × |
| SeaPhi3 mini | 3.8 | Seacom SRL | Ita | | | × |
| SeaPhi3 medium | 14 | Seacom SRL | Ita | | | × |

Questions are crafted and revised by domain experts from CINECA, a non-profit consortium composed of 69 Italian universities, 27 national public research centers, the Italian Ministry of Universities and Research (MUR), and the Italian Ministry of Education (MI). As part of this work, we release the unpublished datasets to the research community. To mitigate the risk of evaluation data contamination when training new LLMs, as highlighted in recent works [24], datasets are

accessible online (hf.co/collections/Detsutut/italian-national-medical-education-pathway-tests) after acknowledging data leakage prevention conditions.

## Experimental setting

Candidate models have been evaluated with a single run over the three datasets. Private models have been accessed solely using their own API or a cloud provider's API (e.g., Amazon Bedrock). Public models below 13B parameters have been hosted locally, while public models above 13B have been hosted on dedicated AWS EC2 instances equipped with 4 GPU A10G Tensor Core. To fit computational constraints, model quantization has been applied for larger models.

In multichoice question-answering, answers are commonly extracted by softmaxing the next token logits of the options, e.g., 'A', 'B', 'C', 'D', and 'E'. However, since logits are unavailable for API-accessed and closed-source LLMs, we relied on regex-based extraction to collect answers from the generated text, ensuring a fair comparison between all the runs. It's worth noting that preliminary explorations have been done on small LLMs comparing the regex-based strategy with two different logit-based strategies. Details about regex and logit exploration are reported in Table C in S1 Text. Given the nature of the task, the temperature parameter has been set to 0.1 to keep text generation grounded. Other hyperparameters have been set with default values.

**Prompt engineering.** Previous studies show that the performance of LLMs might be affected by the format used to build the input prompt, especially for relatively small LLMs [25]. While we acknowledge the importance of prompt engineering, prompt optimization is beyond the scope of our experiment, which focuses on relative comparisons rather than absolute performance. For this reason, we adopted a standardized, near-default prompting setup (i.e., 'vanilla' prompt) to ensure comparability across a large and heterogeneous set of models, including both generalist and specialized ones, with sizes ranging from a few billion to several hundred billion parameters. This approach allowed us to focus on broad performance trends rather than task-specific prompt optimization. The vanilla prompt is reported in Fig 1.

**Evaluation metrics.** Candidate models have been evaluated in terms of accuracy and format accuracy. Accuracy measures the proportion of correct predictions compared to the total number of predictions. It evaluates the model's overall performance in generating the right answers.

```
Rispondi alla seguente domanda a scelta multipla{% if scenario %}, facendo riferimento allo
scenario descritto{% endif %}. Non aggiungere spiegazioni o dettagli.

{% if scenario %}
Scenario: {{ additional_scenario }}
{% endif %}

{{ question_text }}

A) {{ option_a }}
B) {{ option_b }}
C) {{ option_c }}
D) {{ option_d }}
E) {{ option_e }}

{% if not instruct %}
La risposta corretta è
{%endif %}
```

**Fig 1. The vanilla prompt, in Italian language, used for QA in jinja2 format.** In the ITMRA test, some questions refer to a common scenario provided at the beginning of the section, hence the conditional blocks highlighted in blue text. To facilitate properly formatted generation, an additional completion-style string (colored in red in the figure) is added at the end of the prompt only for non-instruct models.

Format Accuracy ensures that the model's outputs follow the required structure, such as selecting one of the valid options (A, B, C, D, E) in a multichoice question. A response may achieve high format accuracy but still be incorrect in content, as format correctness does not imply logical correctness.

Combining accuracy with format checks helps differentiate between validly formatted but incorrect answers and outright invalid outputs. A model with poor format accuracy may need adjustments to its decoding process, such as constrained sampling or vocabulary restriction.

To enable a fair comparison between model performance and average human performance, we also calculated pointwise scores. The pointwise score metric combines positive scoring for correct answers with penalties for incorrect ones, mirroring the structure of real medical exams. The scoring system used to calculate the pointwise score for each exam is reported in Table 3. It is important to note that LLMs have not been evaluated in a setting where questions can be left unanswered, meaning they always provide an answer, which is either correct or incorrect. Participants' scores and other aggregate data regarding national tests are periodically reported by the National Agency for the Assessment of University Systems and Research (ANVUR).

## 3. Results

Given the comprehensive testing of many models with different features across several datasets, the resulting collection of tables and plots is extensive. To effectively communicate the key insights and findings, we have selected the most relevant and illustrative tables and plots. This choice allows us to highlight the most meaningful comparisons, trends, and patterns that emerged from our analysis, ensuring clarity and focus in the presentation of our results. More details are available in Table D1, Figure D, and Table D2 in S1 Text.

Models have been grouped by size (i.e., active parameter count) into 6 categories: <=1B (XS), <=5 (S), <=10 (M), <=50 (L), <=100 (XL), >100 (XXL). Category-wise accuracy is analyzed in Fig 2. Overall, the best-performing model is Gpt 4o, with a total accuracy of 0.913, as reported in Table 4. For each size category, the best performing models are Igea 1B Instruct for XS models, SeaPhi3 mini for S models, Gemini 1.5 Flash 8B and Mixstral 8x7B for M models, Gemini 1.5 Flash and Gpt 4o mini for L models, Claude 3.5 Sonnet for XL models, and Llama 3.1 405B and GPT 4o for XXL models.

Overall, format accuracy is perfect or near-perfect across all model sizes, except for XS models, as shown in Fig 3. It's worth noting that two of the outliers of groups S and M are non-instruct models.

To assess whether the different expected level of expertise of candidates and the complexity of questions affects the models' performances, repeated measures ANOVA test has been performed (n = 37, alpha = 0.05). While ITMLE and ITMRA are strictly medical, ITMSA also includes questions not directly related to the healthcare domain. For this reason, we also tested two additional versions of ITMSA in the comparison, considering STEM-related questions only (i.e., removing general knowledge questions) and biochemistry-related questions (i.e., also removing logic, physics, and maths questions).

### Medical school admission test (ITMSA)

For the ITMSA dataset, we inspected topic-wise accuracy over the top-performing models of each category (Fig 4) and calculated the equivalent test score for each model comparing it with the average performance of human candidates and

**Table 3. Scoring system used in the national exams.**

| Test | Correct Answer | Unanswered | Wrong Answer | Max Score |
|---|---|---|---|---|
| Medical School (ITMSA) | +1.5 | 0 | -0.4 | 90 |
| Residency (ITMRA) | +1 | 0 | -0.25 | 140 |
| Medical License (ITMLE) | +1 | 0 | -0.25 | 180 |

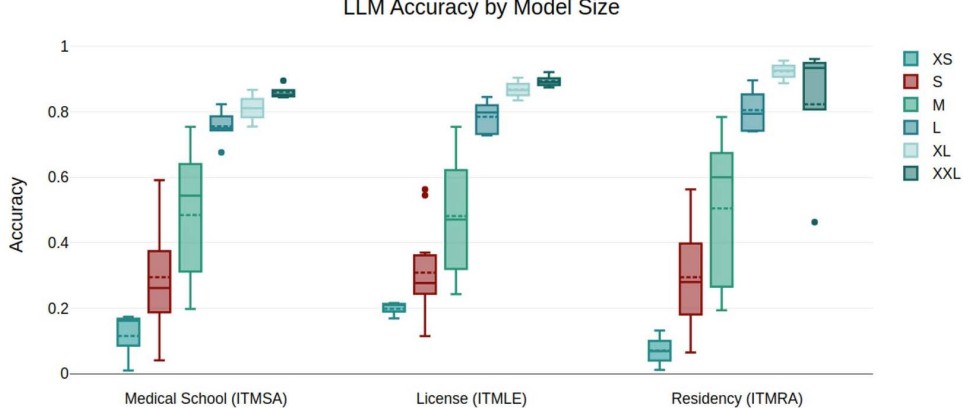

**Fig 2. Boxplot analysis of accuracy among models, grouped by size category, over different datasets.**

**Table 4. Average and maximum (bracketed) accuracy for different model size categories over the different datasets. For each group and each dataset, the name of the best-performing model is reported.**

| Size | Placement (ITMSA) | | Medical License (ITMLE) | | Residency (ITMRA) | | Overall | |
|---|---|---|---|---|---|---|---|---|
| | Average (Max) | Best Model | Average (Max) | Best Model | Average (Max) | Best Model | Average (Max) | Best Model |
| XS | 0.115 (0.174) | Igea 1B Instruct | 0.199 (0.216) | Igea 1B Instruct | 0.071 (0.132) | Apollo 0.5B | 0.128 (0.154) | Apollo 0.5B |
| S | 0.295 (0.591) | SeaPhi3 mini | 0.309 (0.563) | SeaPhi3 mini | 0.307 (0.563) | SeaPhi3 mini | 0.303 (0.572) | SeaPhi3 mini |
| M | 0.465 (0.754) | Gemini 1.5 Flash 8B | 0.464 (0.754) | Gemini 1.5 Flash 8B | 0.491 (0.784) | Mixstral 8x7B Instruct | 0.473 (0.762) | Gemini 1.5 Flash 8B |
| L | 0.756 (0.823) | Gemini 1.5 Flash | 0.785 (0.845) | Gpt 4o mini | 0.805 (0.896) | Gpt 4o mini | 0.782 (0.842) | Gpt 4o mini |
| XL | 0.811 (0.867) | Claude 3.5 Sonnet | 0.868 (0.904) | Claude 3.5 Sonnet | **0.923 (0.956)** | Claude 3.5 Sonnet | **0.868 (0.909)** | Claude 3.5 Sonnet |
| XXL | **0.861 (0.895)** | Llama 3.1 405B Instruct | **0.894 (0.921)** | Gpt 4o | 0.823 (0.961) | Gpt 4o | 0.859 (0.913) | Gpt 4o |

the actual pass threshold (Fig 5). The pass threshold (before repechage) varies each year due to differences in the number of test participants, the difficulty level of the test, and the enrollment capacity at universities nationwide. To account for this variability, the results have also been compared with the minimum, maximum, and average pass thresholds recorded since 2018 (and up to 2024).

## Medical licensing examination: ITMLE

As illustrated in Fig 6, we inspected the maximum achievable accuracy for each model size on the medical license test, differentiating between Italian and Multilingual models and inspecting the trends with two logarithmic regressions ($R^2$ = 0.769 for Italian, $R^2$ = 0.692 for Multilingual), reporting the pass threshold and the baseline accuracy obtained with random guessing. No data about the average human performance was available for this test. Given the large number of clinical topics covered by the 6696 questions of ITMLE, we also conducted a more focused evaluation of Llama 3 8B and its language-specific and domain-specific derivations. Thanks to its strong performance, permissive license, and relatively

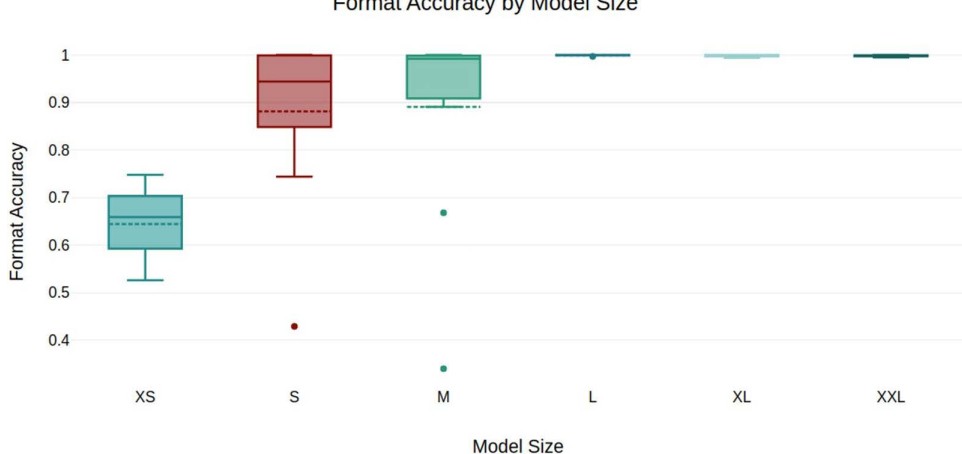

**Fig 3. Format Accuracy by Model Size.** Outliers: Minerva 3B (S), Meditron 3 8B (M), Gemma 9B (M).

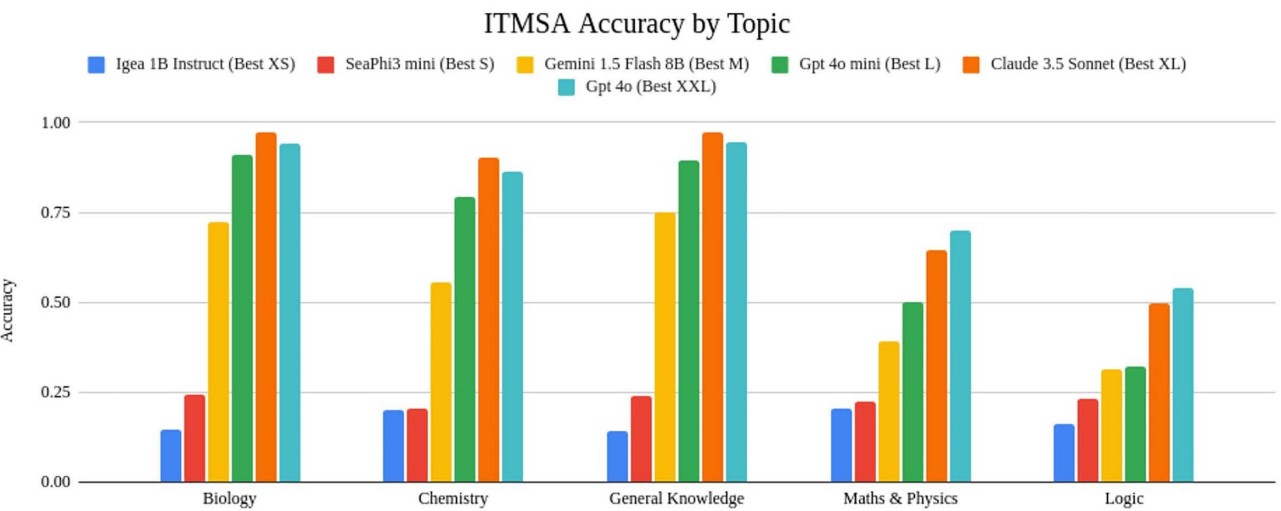

**Fig 4. Accuracy of the best model for each size category on ITMSA, divided by topic.**

small size that makes it accessible for training, Llama 3 8B has become one of the most widely adapted and customized language models. On the language axis, we compared both the original Llama with its Italian derivate and the original Qwen with its Italian counterpart. On the domain axis, we compared the original Llama with the three medical fine-tunings of Llama included in this evaluation, namely MedLlama, Meditron, and MedX. These comparisons are shown in detail in Fig 7, where accuracies are reported for each medical field tested in the medical licensing examination.

### Residency admission test: ITMRA

When scaling large language models, the dynamics of how specific answers change with model size is often understudied. Insights into how scaling helps turning incorrect answers into correct ones for Llama 3.1 in ITMRA are illustrated in Fig 8. The data indicates a clear trend of improvement: as the model size grows, the number of correct answers increases

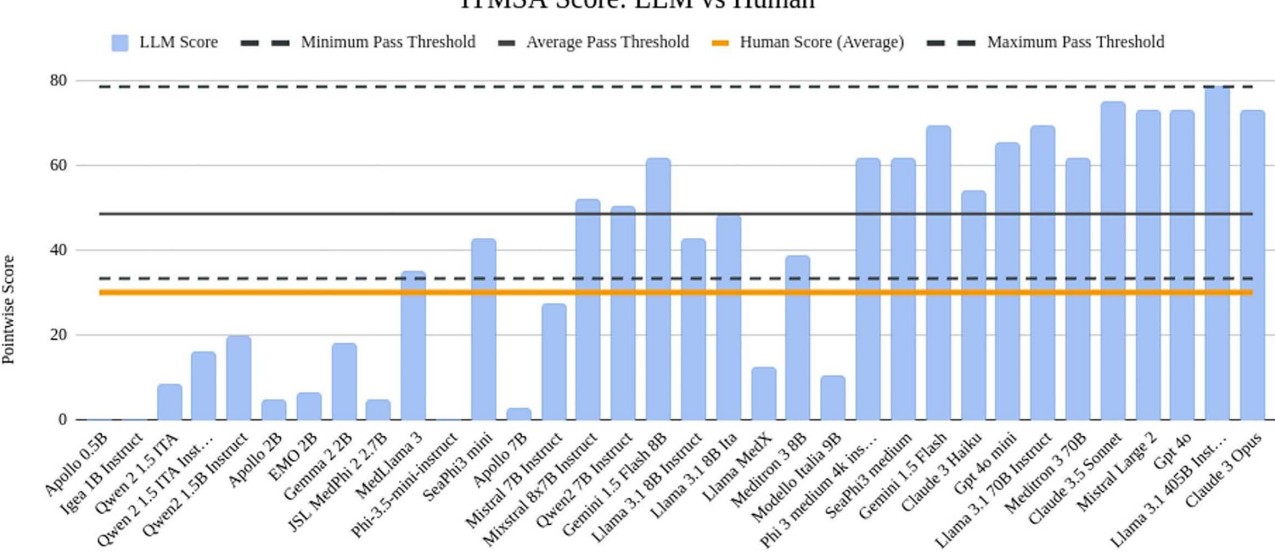

**Fig 5. Placement test scores of all LLMs compared with the average scores of human participants and the thresholds required for admission to the least competitive medical school in the country (minimum, average, and maximum) before repechage.** Models on the X axis have been ordered by size.

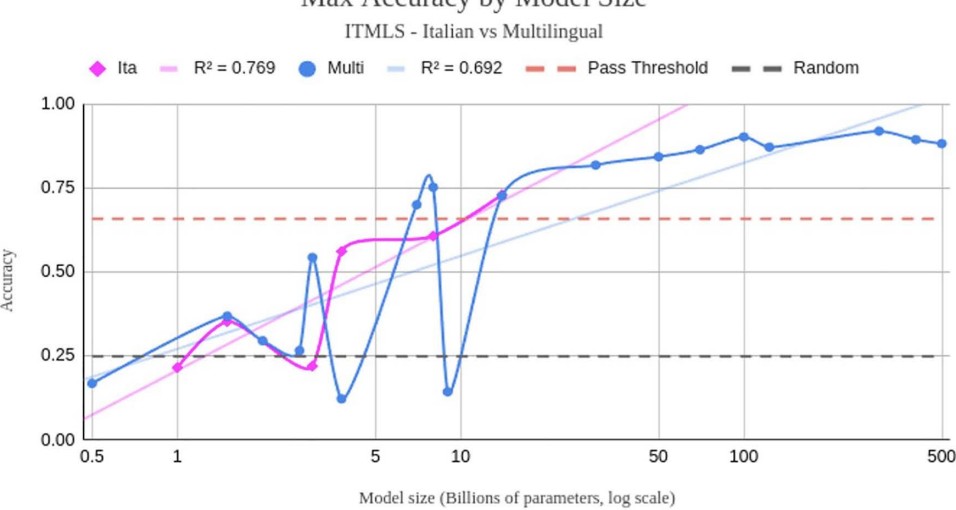

**Fig 6. Max accuracy achieved by LLMs on the ITMLE dataset, divided by Italian (blue) models and Multilingual (red) models, along with their logarithmic trendlines.** Pass and random guessing accuracy thresholds are reported respectively in green and yellow.

noticeably, from 604 in the smallest model (8B) to 912 in the largest (405B). The chart also highlights transitions between correct and incorrect answers; some previously incorrect answers turn into correct as the model scales, while a smaller proportion of correct answers roll back to being incorrect. For instance, at size L (70B) in the middle of the figure, most answers have improved compared to the smallest model, but a small portion of correct responses has been misclassified.

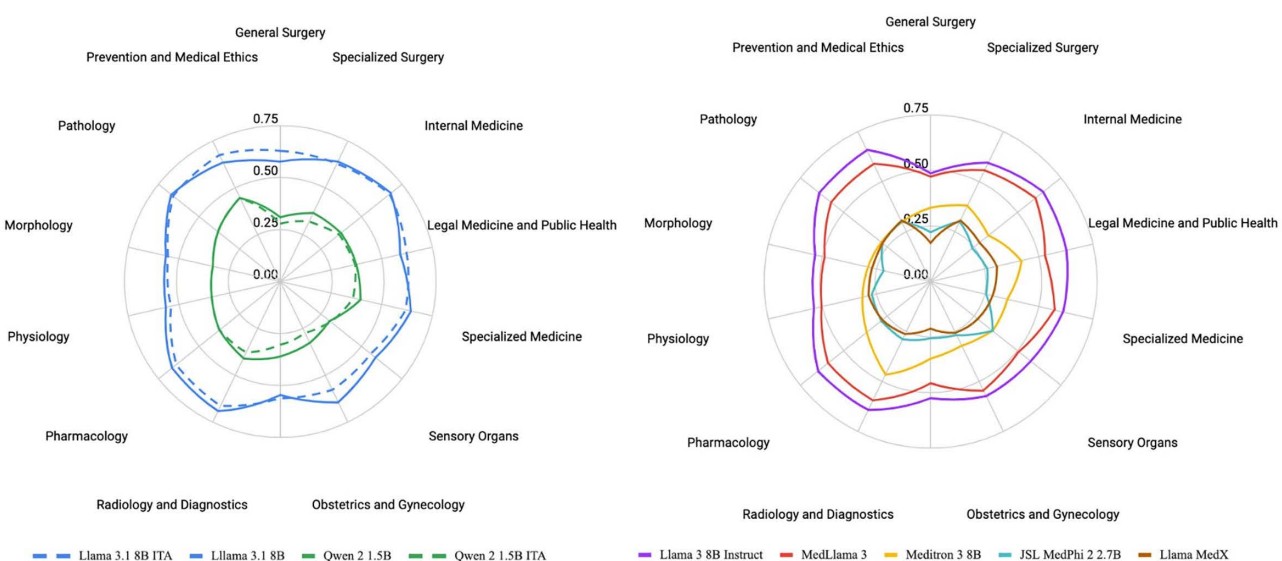

**Fig 7. Topic-wise accuracy across selected multilingual-Italian model pairs (left) and general-purpose Llama 3.1 vs its medical counterparts (right).**

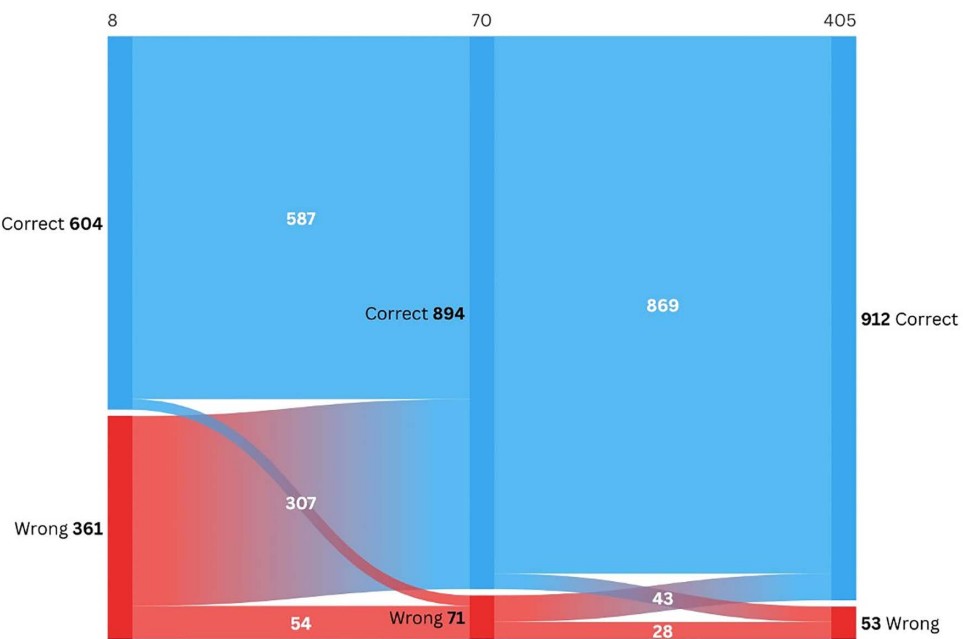

**Fig 8. Accuracy progression of Llama 3.1 versions on the ITMRA dataset as model size increases.** The x-axis represents different model sizes, while the y-axis categorizes answers as correct (blue) or wrong (red).

Nineteen answers stay wrong despite the Llama model size. Among these answers, four were answered incorrectly also by all the other top-performing models. This error analysis is thoroughly explored in Table E in S1 Text.

Unlike previously analyzed datasets, ITMRA includes information on results obtained for each year (2017–2023), providing a clearly defined temporal dimension. This allows for tracking performance changes over time, enabling us to analyze performance trends for both human participants and large language models (LLMs) across multiple years. These trends are illustrated in Fig 9, offering a comparative view of how performance evolves for both groups.

## 4. Discussion

### Model size matters

The study shows a consistent improvement in performance as model size increases for expert-curated medical question answering in the Italian language, which aligns with broader findings in the field of NLP, where scaling up models has led to state-of-the-art results across various tasks, domains, and languages. Interestingly, our results reveal no substantial difference between the performance of XXL models (greater than 100 billion parameters) and XL models (approximately 50–100 billion parameters). This plateau in performance suggests diminishing returns as parameter counts grow beyond a certain threshold. From a practical perspective, this finding implies little or no need to further increase model size beyond 100 billion active parameters for the multiple-choice medical Q&A task evaluated in this study. These results are consistent with observations in other domains and tasks, where excessively large models can introduce additional complexity and inefficiencies without yielding proportionate improvements. Task saturation could contribute to the observed performance plateau, as many items are easily solved by high-performing models, reducing their ability to differentiate top-tier systems. However, we also observed that specific examples correctly answered by the 8B and 70B versions of LLaMA were sometimes missed by the 405B version, and vice versa. While this analysis is limited to a single model family, it suggests that the plateau is more indicative of intrinsic scaling effects rather than task saturation.

Furthermore, XS language models were found to be largely ineffective, showing remarkably low accuracy and inability to consistently adhere to format constraints, making them unsuitable for meaningful applications. In contrast, XL and XXL models exhibit remarkable improvements in both task accuracy and format adherence. This underscores the limitations of smaller models in complex, knowledge-intensive tasks and reinforces the importance of scale for achieving competitive performance.

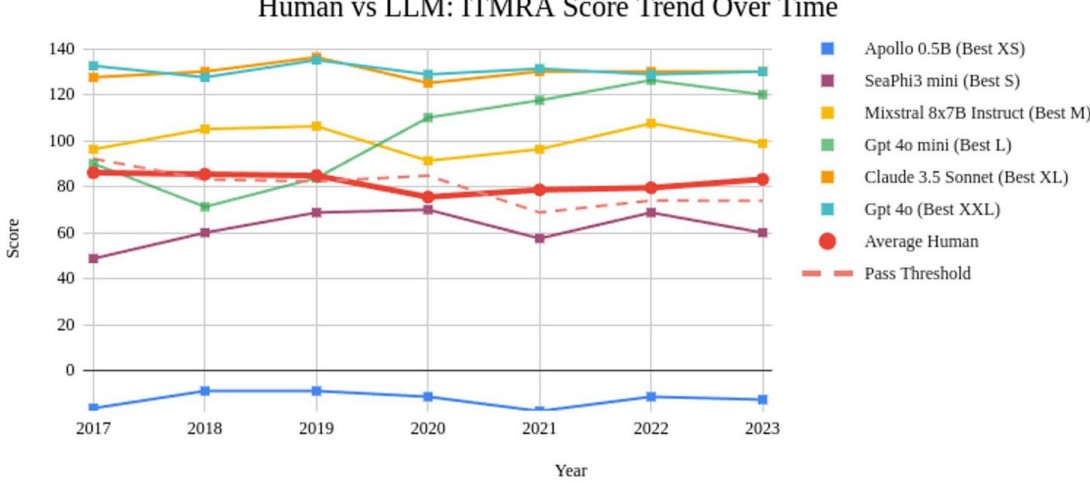

**Fig 9. Residency admission pointwise score for the best LLM for each size category compared with the average score of human participants over the years.** Negative scores (e.g., Apollo 0.5B) are possible since wrong answers are penalized with -0.25 points.

Interestingly, near-perfect format accuracy is achieved by models larger than or equal to the L-size group (i.e., > 10B). This observation highlights the intrinsic ability of LLMs to conform to task-specific formats when adequately scaled, reducing the need for additional post-processing of the LLM outputs to enforce format adherence. This capability has practical implications for applications that rely on structured output, such as automated test scoring or content generation.

## Domain alignment and language localization may be superfluous

One of the most relevant findings of this research is the lack of notable performance improvement from fine-tuning on domain-specific data. This finding challenges the prevailing assumption that domain-specific models are inherently superior for specialized tasks [26]. The generalization ability of the latest large language models appears sufficient to match or even exceed the zero-shot performance of domain-specific models, even without any particular prompt engineering or task-specific adaptation. These results underscore the versatility of generic LLMs and suggest that fine-tuning on domain-specific data may not be necessary anymore in tasks like multiple-choice Q&A. For medical institutions or organizations, this translates to a reduced overhead in developing and maintaining their own specialized models.

Similarly, fine-tuning models on language-specific data seems to have a limited impact on performance in the evaluated setting. While modest improvements were observed in very small models (XS and S), the benefits diminish as models grow larger. This trend indicates that larger multi-lingual models inherently capture sufficient linguistic nuances to handle language-specific tasks effectively, even without additional localization.

Interestingly, recent findings in cognitive science offer a compelling parallel: superior clinical reasoning may arise not from isolated medical knowledge alone, but from the integration of broad general knowledge and experience with domain-specific medical training [27]. While clinical cognition in AI systems is clearly distinct from human cognition [28], this analogy is consistent with growing evidence that larger, general-purpose language models can exhibit stronger reasoning capabilities than more specialized models, despite relying on comparatively less domain-specific training data.

Due to the absence of single-language models exceeding 9 billion parameters in our study, it's worth noting that we cannot definitively conclude how such models might perform at larger scales (M, L, and beyond). This limitation highlights a potential avenue for future research, especially for languages where large monolingual models are still scarce, provided the amount of data needed to train them is indeed sourceable in such low-resource languages.

## Medical complexity of the question does not make it "harder" for LLMs

No significant difference (p = 0.3238) in accuracy was observed between the different datasets, independently from the level of expertise and knowledge of medicine they have been designed to test. This reflects the recall-oriented nature of LLMs, which fits well with medical education and assessments that often emphasize the ability to retrieve factual knowledge efficiently. However, there is a statistical difference (p < 0.00001) in performance between STEM-only, biochemistry-only, and full ITMSA, primarily driven by questions related to logic and mathematics, where LLMs struggle more [29]. Logical and mathematical reasoning demands higher-order cognitive skills that remain challenging for LLMs, highlighting the importance of enhancing reasoning capabilities [30], as these areas are critical for tasks that go beyond rote memorization. Educational institutions and test designers could leverage this insight to identify and address specific areas where learning gaps are most pronounced, as being recently done in simulated medical education settings [31].

## Human vs LLM: big models have an edge

Models with around 7 billion parameters perform comparably to the average human participant on medical assessments such as ITMRA and ITMSA. In contrast, models exceeding 50 billion parameters demonstrate superior capabilities, consistently outperforming human participants in answering multiple-choice questions. As shown in Fig 10, this trend peaks with very large LLMs like GPT-4o and Llama 3.1 405B exceeding average human performance by over 30 points

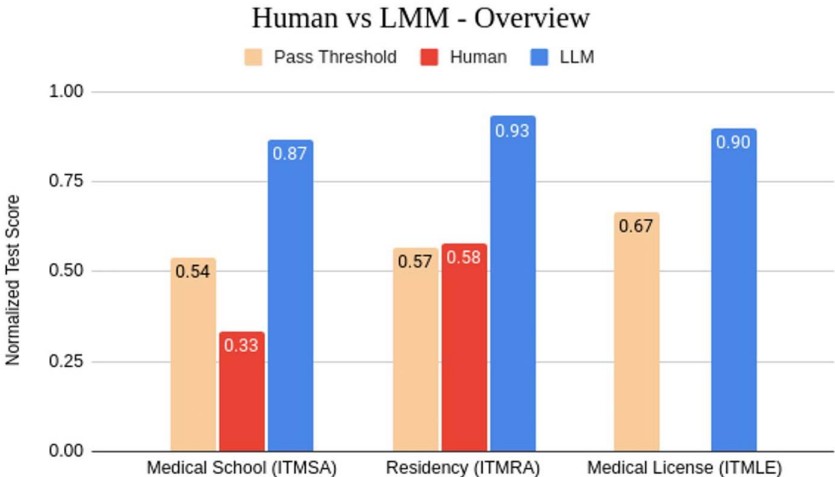

**Fig 10. Comparison between human participants (average performance) and best LLM in terms of normalized pointwise scores, along with the average pass threshold.** Best model for ITMSA: Llama 3.1 405B Instruct. Best model for ITMRA and ITMLE: GPT-4o. There is no available data on candidates participating in medical licensing examinations.

on average, acing the corresponding national assessments. This achievement is particularly noteworthy given that these models lack access to these questions of our datasets, unpublished before the cutoff date.

## 5. Conclusion

In this study, we performed a comparative evaluation of multilingual and Italian-specific models, as well as general-purpose and healthcare-specific models, spanning both open-source and proprietary architectures of varying sizes. The evaluation has been conducted over official unpublished-before Italian medical multichoice Q&A tests, varying in topics and complexity.

Our results demonstrate that while recent advancements in attention mechanisms, compression strategies, and optimization techniques have garnered attention, they have not diminished the importance of scaling model size, with larger models consistently exceeding the pass threshold and outperforming human participants across all tests, with the best model (i.e., GPT-4o) achieving over 96% accuracy on residency admission tests and steadily performing over 90% accuracy overall. It's also worth noting that the performance gap between open-weight and closed-weight models is narrow, in particular when comparing large, state-of-the-art models like Llama 3.1 405B or GPT-4o. This narrowing performance gap aligns with the trend observed in popular English benchmarks [30].

The lack of relevant performance improvement from domain alignment and single-language localization is a particularly relevant discovery. Such findings challenge the assumption that domain-specific models are inherently superior for specialized tasks, underscoring the versatility of modern generic LLMs and suggesting that fine-tuning on domain-specific data may not be necessary anymore even when dealing with highly specialized scenarios like medical assessments.

For medical institutions and organizations, this could influence development strategies, making it more effective to invest in deploying larger, general-purpose, multilingual models rather than dedicating massive resources to collect the appropriate data for specialized, single-language training and develop custom domain-specific models. For those institutions wanting to adopt large general-purpose models in real-world deployment, the FUTURE-AI structured framework [32] represents a valid resource to consider important safety aspects like Issues of bias, hallucinations, data privacy, and guideline alignment, proposing a set of concrete safeguards and operational practices that go beyond performance metrics.

Our study faced several constraints that are worth discussing. With an ever-growing pool of available models and configurations, it is impossible to test every combination exhaustively. Some models might have been evaluated under suboptimal conditions, and the best-performing candidate might have been filtered out a priori when applying the selection criteria.

Our analysis was also restricted to text-based scenarios, excluding questions requiring clinical image inspection. This choice limits the scope of our findings, particularly for tasks that rely on visual information. The evaluation also focused solely on single-round multiple-choice question-answering. While this provides insights into a specific task, clinical practice encompasses a much broader set of activities, including multi-step decision-making and diverse task types, which remain outside the scope of this study. By including closed-source models for a more comprehensive comparison, we also have to restrict our observations to textual outputs, as we were unable to access weights and logits for all the candidates.

Moreover, the question-answering paradigm primarily measures factual recall, which captures only a fraction of the competencies needed in medical practice. LLM performance on more complex reasoning tasks (e.g., differential diagnosis) and conversational settings (e.g., simulated patient-doctor dialogues) remains inadequate and markedly lower than in QA scenarios [33,34]. This preliminary study serves as a foundational step toward deeper investigations in this direction.

Building on the current findings, we identify several directions for advancing future research. One promising direction involves the development of an automated benchmarking framework tailored specifically for official Italian medical assessments. Such a framework would facilitate the streamlined testing and evaluation of emerging models, ensuring a more efficient and standardized approach to model comparison. Another critical extension lies in exploring models capable of processing multimodal data, integrating both textual and visual inputs providing a more comprehensive evaluation of the assessments. Furthermore, expanding the scope of analysis to encompass nursing national assessments would offer a broader perspective on clinical practice, allowing for a more comprehensive evaluation across diverse professional domains within healthcare. Finally, with widely available generic models consistently performing above threshold, and often surpassing the average performance of humans taking the test, we advocate that further assessments of medical models should move out of the labs and closer to actual implementation scenarios, away from multiple-choice Q&A benchmarks. Aside from pure performance, other key factors are likely to play an important part in model selection and its uptake in the medical field: intricacies of interacting in natural language with healthcare professionals and patients; importantly the integration of such tools in healthcare delivery processes; alignment with local and international clinical practice guidelines, and its ability to align with physician and patient preferences; and technical and semantic interoperability with other pre-existing health information systems. In the medical domain, thorough evaluation of LLMs is still rare and fragmented [35] and, given the good performance that such models attain in baseline Q&A benchmarks including the one described in this paper, prospective and human-based evaluation on real-world medical cases is the natural next step for localized- medical-specific models.

In parallel with these research trajectories, it is worth acknowledging that domain-specific models may still hold value when positioned within agentic-AI architectures rather than as standalone solutions. In this configuration, a large general-purpose LLM operates as an orchestrator: interpreting user intent and routing complex requests to small models fine-tuned for specialized tasks. The synergy between general and specialized models is beneficial for structured clinical reasoning, guideline-aligned processes, or large-scale curated medical knowledge [36,37]. Acknowledging such architectures reconciles our findings with ongoing domain-specific efforts, positioning generic and specialized approaches as complementary components within broader medical AI systems.

## Supporting information

**S1 Text. Supporting Information.** Table A: The TRIPOD-LLM Checklist. Table B1: Accuracy and Format Accuracy (FA) for all the models tested. Table B2: Reference endpoints and repositories for each model included in the study. Table C: Comparison between regex-based and logit-based mode in terms of accuracy, format accuracy and relative accuracy

change with respect to the regex-based method. Table D1: Results for all the datasets and all the models in terms of accuracy and format accuracy. Figure D: Accuracy achieved by LLMs on the ITMLE, ITMRA and ITMSA datasets, along with the average value and logarithmic trendline. Diamond-shaped points indicate pre-graduate examination results, while star-shaped points indicate post-graduate examination results. Table D2: Maximum value, average value, and variance for each test and each size group. Table E: Subset of questions answered incorrectly by all the top-performing LLMs tested. The table reports the original question in Italian (correct option highlighted), the answer chosen most frequently by top LLMs, the average estimated difficulty, the average agreement with the highlighted correct answer, and the average agreement with the answer chosen by LLMs.
(DOCX)

## Author contributions

**Conceptualization:** Tommaso Mario Buonocore, Enea Parimbelli.

**Data curation:** Tommaso Mario Buonocore, Antonio Russo, Dario Mingarelli.

**Formal analysis:** Tommaso Mario Buonocore.

**Investigation:** Tommaso Mario Buonocore.

**Methodology:** Tommaso Mario Buonocore.

**Project administration:** Tommaso Mario Buonocore, Enea Parimbelli.

**Resources:** Tommaso Mario Buonocore, Antonio Russo, Dario Mingarelli.

**Software:** Tommaso Mario Buonocore.

**Supervision:** Tommaso Mario Buonocore, Enea Parimbelli.

**Validation:** Tommaso Mario Buonocore, Enea Parimbelli.

**Visualization:** Tommaso Mario Buonocore.

**Writing – original draft:** Tommaso Mario Buonocore, Enea Parimbelli.

**Writing – review & editing:** Tommaso Mario Buonocore, Dario Mingarelli, Enea Parimbelli.

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
