## [Decision Letter · Decision Letter 0]

28 Oct 2025

Response to Reviewers'. This file does not need to include responses to any formatting updates and technical items listed in the 'Journal Requirements' section below.'. This file does not need to include responses to any formatting updates and technical items listed in the 'Journal Requirements' section below.* A marked-up copy of your manuscript that highlights changes made to the original version. You should upload this as a separate file labeled 'Revised Manuscript with Track Changes'.'.* An unmarked version of your revised paper without tracked changes. You should upload this as a separate file labeled 'Manuscript'.'. If you would like to make changes to your financial disclosure, competing interests statement, or data availability statement, please make these updates within the submission form at the time of resubmission. Guidelines for resubmitting your figure files are available below the reviewer comments at the end of this letter. We look forward to receiving your revised manuscript. Kind regards, Fatih Uysal, Ph.D.Academic EditorPLOS Digital Health Fatih UysalAcademic EditorPLOS Digital Health Leo Anthony CeliEditor-in-ChiefPLOS Digital Healthorcid.org/0000-0001-6712-6626 **Journal Requirements:**

1. We ask that a manuscript source file is provided at Revision. Please upload your manuscript file as a .doc, .docx, .rtf or .tex.

2. Please upload separate figure files in .tif or .eps format. Also, remove the figures from your manuscript file but keep the legends.

3. We noticed that you used "unpublished data" in the manuscript. We do not allow these references, as the PLOS data access policy requires that all data be either published with the manuscript or made available in a publicly accessible database. Please amend the supplementary material to include the referenced data or remove the references.

4. Please provide an Author Summary. This should appear in your manuscript between the Abstract (if applicable) and the Introduction, and should be 150–200 words long. The aim should be to make your findings accessible to a wide audience that includes both scientists and non-scientists. Sample summaries can be found on our website under Submission Guidelines:

https://journals.plos.org/digitalhealth/s/submission-guidelines#loc-parts-of-a-submission

5. We have noticed that you have uploaded Supporting Information files, but you have not included a list of legends. Please add a full list of legends for your Supporting Information files after the references list.

6. In the online submission form, you indicated that “The datasets analyzed during the current study are not publicly available due to the concrete risk of being scraped by automatic LLM crawlers, compromising the evaluation of new LLMs. However, the datasets can be obtained from the corresponding author upon reasonable request, subject to agreements ensuring adherence to data leakage prevention measures.”.

3. Uploaded as supplementary information.

**Additional Editor Comments (if provided):** Please revise your paper according to the reviewer comments.**Reviewers' Comments:** Reviewer's Responses to Questions

**Comments to the Author**

1. Does this manuscript meet PLOS Digital Health’s publication criteria? Is the manuscript technically sound, and do the data support the conclusions? The manuscript must describe methodologically and ethically rigorous research with conclusions that are appropriately drawn based on the data presented.? Is the manuscript technically sound, and do the data support the conclusions? The manuscript must describe methodologically and ethically rigorous research with conclusions that are appropriately drawn based on the data presented.

Reviewer #1: Yes

Reviewer #2: Partly

2. Has the statistical analysis been performed appropriately and rigorously?

Reviewer #1: Yes

Reviewer #2: No

3. Have the authors made all data underlying the findings in their manuscript fully available (please refer to the Data Availability Statement at the start of the manuscript PDF file)?

The PLOS Data policy requires authors to make all data underlying the findings described in their manuscript fully available without restriction, with rare exception. The data should be provided as part of the manuscript or its supporting information, or deposited to a public repository. For example, in addition to summary statistics, the data points behind means, medians and variance measures should be available. If there are restrictions on publicly sharing data—e.g. participant privacy or use of data from a third party—those must be specified.requires authors to make all data underlying the findings described in their manuscript fully available without restriction, with rare exception. The data should be provided as part of the manuscript or its supporting information, or deposited to a public repository. For example, in addition to summary statistics, the data points behind means, medians and variance measures should be available. If there are restrictions on publicly sharing data—e.g. participant privacy or use of data from a third party—those must be specified.

Reviewer #1: Yes

Reviewer #2: No

4. Is the manuscript presented in an intelligible fashion and written in standard English?

Reviewer #1: Yes

Reviewer #2: Yes

Reviewer #1: It is an important question in medical AI: whether general-purpose multilingual LLMs can match or surpass domain- or language-specialized models in healthcare education.

While the study contributes to the debate on specialization versus generalization, I see areas where it could be strengthened.

1. The conclusion that model size is the main driver of performance, while domain- and language-specific fine-tuning adds little, is consistent with earlier work. Thus, what unique insights does the Italian context add that have not already been demonstrated elsewhere?

2. The evaluation is limited to multiple-choice Q&A, which measures factual recall but not reasoning, contextual judgment, or patient communication. This narrows the conclusions that can be drawn about clinical usefulness. Why was the evaluation restricted to Q&A, and how do the authors see the results generalizing to reasoning-based or multimodal tasks?

3. Could prompt design or few-shot prompting alter the relative performance of generalist versus specialized models?

The inclusion of reasoning tasks, multimodal inputs, or retrieval-augmented generation (RAG) would require appropriate citations to comparable studies for context.

The discussion notes that large multilingual models reduce the need for localization, but practical deployment requires more than accuracy. Issues of bias, hallucinations, data privacy, and guideline alignment are crucial but not fully addressed. What safeguards are proposed for these risks if institutions adopt general-purpose models?

The authors should also cite work addressing reliability, bias, and clinical integration to support these points.

For improvement, I suggest:

1. Positioning the study more clearly against existing benchmarks (e.g., ChatGPT: transforming healthcare with AI, Retrieval-Augmented Generation (RAG) in Healthcare: A Comprehensive Review).

2. Broadening evaluation beyond factual Q&A to include reasoning or multimodal tasks.

3. Reflecting on where domain-specific models may still be valuable, with reference to supporting literature.

4. Expanding discussion of implementation aspects such as reliability, integration with health systems, and guideline alignment, again supported by citations.

Finally, since hybrid approaches like RAG are increasingly seen as a middle ground, the paper would benefit from situating its findings in this context. Would retrieval-enhanced multilingual models shift the conclusion that domain-specific training is unnecessary?

More broadly, how can exam-level results be translated into real-world clinical reliability, where reasoning, context, and patient interaction are central?

Reviewer #2: The paper investigates whether generalist, multilingual LLMs outperform specialized, domain-specific, or language-localized models in the Italian medical education domain. Using three unpublished multiple-choice exam datasets (undergraduate, licensing, and residency), the authors benchmarked 37 models ranging from <1B to >400B parameters. Results suggest that model size—not domain or linguistic specialization—is the key performance determinant. Larger general-purpose models (e.g., GPT-4o, Llama 3.1 405B) surpass human-level accuracy on these tasks, leading to the conclusion that domain-specific pretraining may be increasingly redundant.

Even tough the paper address some relevant question ( whether domain specialization still provides measurable benefits in the age of foundation models), the dataset used is, as far as I know, innovative, the inclusion of many LLM is to be appreciated and the manuscript is clear; some critical aspects should be taken into account:

1. Methodology

- Prompting consistency and fairness:

The use of a single “vanilla” prompt, could the authors quantify the potential bias (e.g., sensitivity tests with 1–2 optimized prompts per model class).

- Single-run evaluation:

Only one run per model is reported, despite known stochastic variability even with low temperature (0.1). Bootstrapped or repeated runs would strengthen claims of statistical significance, particularly given small performance gaps among high-end models.

- Different inference modalities:

Some models were accessed locally, others through APIs. Latency, tokenization, and decoding parameters may differ, potentially affecting accuracy and output formatting. The fairness of comparing closed APIs (e.g., GPT-4o) to local quantized models should be further justified.

- Regex-based answer extraction:

The regex-based parsing used for closed models may introduce systematic extraction errors. Although supplementary materials mention validation, no quantitative estimate of extraction reliability is reported.

2. Statistical analysis

- The ANOVA test on dataset difficulty (p = 0.3238) is informative but insufficient. Confidence intervals for model accuracy, effect sizes, or pairwise post-hoc comparisons are not presented. Without them, it is difficult to assess the robustness of claims like “no measurable advantage from specialization.”

- No correction for multiple comparisons is described, even though many models and datasets were tested.

- The human–LLM comparison lacks sample sizes and standard deviations for human results, making it hard to contextualize the “30-point” advantage.

3.Interpretation of results

- In my opinion the conclusion that domain- or language-specific training is “unnecessary” is overstated. The task (multiple-choice Q&A) primarily measures factual recall, not reasoning, narrative understanding, or generation quality. Evidence from other domains (e.g., clinical summarization, report generation, guideline reasoning) still supports the utility of domain adaptation.

- What about the plateau beyond 100 B parameters ? can it be related to a task saturation rather than intrinsic scaling limits.

- The conceptual distinction between the concepts of “domain specialization” and “optimized adaptation to a single task” is not entirely clear.

4. Dataset transparency and availability

- The decision not to release datasets publicly for "to prevent LLM crawling" is unconventional and undermines the reproducibility of the central claims. Given that the results hinge on proprietary exam data, a redacted or paraphrased subset could still be shared for verification.

- Dataset composition (e.g., question length distribution, vocabulary overlap with pretraining corpora) is not analyzed. Possible contamination—although claimed to be minimal—is not empirically tested (e.g., via n-gram overlap with known training corpora).

5. Novelty vs. existing literature

- Similar analyses exist for multilingual LLMs in medical contexts (e.g., Alfertshofer et al., 2024; Lehman et al., 2023; Riedemann et al., 2024). The novelty here lies in the Italian setting, but the paper does not sufficiently distinguish its methodological contribution from prior multilingual medical-exam benchmarks.

- The link between cognitive-science analogies and LLM generalization (Introduction, pp. 2–3) is interesting but speculative; it would benefit from quantitative backing or omission.

6. Reproducibility and Ethical Compliance

- Reproducibility is partially satisfied (GitHub link provided), but local scripts alone are insufficient without data access.

**Do you want your identity to be public for this peer review?** For information about this choice, including consent withdrawal, please see our Privacy Policy..

Reviewer #1: No

Reviewer #2: No

**Figure resubmission:**  While revising your submission, we strongly recommend that you use PLOS’s NAAS tool (https://ngplosjournals.pagemajik.ai/artanalysis) to test your figure files. NAAS can convert your figure files to the TIFF file type and meet basic requirements (such as print size, resolution), or provide you with a report on issues that do not meet our requirements and that NAAS cannot fix. 

**Reproducibility:** To enhance the reproducibility of your results, we recommend that authors of applicable studies deposit laboratory protocols in protocols.io, where a protocol can be assigned its own identifier (DOI) such that it can be cited independently in the future. Additionally, PLOS ONE offers an option to publish peer-reviewed clinical study protocols. Read more information on sharing protocols at https://plos.org/protocols?utm_medium=editorial-email&utm_source=authorletters&utm_campaign=protocols To enhance the reproducibility of your results, we recommend that authors of applicable studies deposit laboratory protocols in protocols.io, where a protocol can be assigned its own identifier (DOI) such that it can be cited independently in the future. Additionally, PLOS ONE offers an option to publish peer-reviewed clinical study protocols. Read more information on sharing protocols at https://plos.org/protocols?utm_medium=editorial-email&utm_source=authorletters&utm_campaign=protocols

---

## [Decision Letter · Decision Letter 1]

30 Mar 2026

Generalist Large Language Models in a Specialized World: Evidence from the Italian National Medical Education Pathway

PDIG-D-25-00263R1

Dear Mr. Buonocore,

We are pleased to inform you that your manuscript 'Generalist Large Language Models in a Specialized World: Evidence from the Italian National Medical Education Pathway' has been provisionally accepted for publication in PLOS Digital Health.

Best regards,

Fatih Uysal, Ph.D.

Academic Editor

PLOS Digital Health

**Additional Editor Comments (if provided):**

Considering the final version of the paper, it is observed to be sufficient.

**Reviewer Comments (if any, and for reference):**

Reviewer's Responses to Questions

**Comments to the Author**

Reviewer #1: All comments have been addressed

publication criteria? Is the manuscript technically sound, and do the data support the conclusions? The manuscript must describe methodologically and ethically rigorous research with conclusions that are appropriately drawn based on the data presented.? Is the manuscript technically sound, and do the data support the conclusions? The manuscript must describe methodologically and ethically rigorous research with conclusions that are appropriately drawn based on the data presented.

Reviewer #1: Yes

3. Has the statistical analysis been performed appropriately and rigorously?

Reviewer #1: Yes

4. Have the authors made all data underlying the findings in their manuscript fully available (please refer to the Data Availability Statement at the start of the manuscript PDF file)?

The PLOS Data policy requires authors to make all data underlying the findings described in their manuscript fully available without restriction, with rare exception. The data should be provided as part of the manuscript or its supporting information, or deposited to a public repository. For example, in addition to summary statistics, the data points behind means, medians and variance measures should be available. If there are restrictions on publicly sharing data—e.g. participant privacy or use of data from a third party—those must be specified.requires authors to make all data underlying the findings described in their manuscript fully available without restriction, with rare exception. The data should be provided as part of the manuscript or its supporting information, or deposited to a public repository. For example, in addition to summary statistics, the data points behind means, medians and variance measures should be available. If there are restrictions on publicly sharing data—e.g. participant privacy or use of data from a third party—those must be specified.

Reviewer #1: Yes

5. Is the manuscript presented in an intelligible fashion and written in standard English?

Reviewer #1: Yes

Reviewer #1: Good work.

**Do you want your identity to be public for this peer review?** For information about this choice, including consent withdrawal, please see our Privacy Policy..

Reviewer #1: No
